# Predicting Food–Drug Interactions between Piperine and CYP3A4 Substrate Drugs Using PBPK Modeling

**DOI:** 10.3390/ijms252010955

**Published:** 2024-10-11

**Authors:** Feifei Lin, Yingchun Hu, Yifan Zhang, Lijuan Zhao, Dafang Zhong, Jia Liu

**Affiliations:** 1School of Pharmaceutical Science and Technology, Hangzhou Institute for Advanced Study, University of Chinese Academy of Sciences, Hangzhou 310058, China; feifei.lin@simm.ac.cn; 2Shanghai Institute of Materia Medica, Chinese Academy of Sciences, Shanghai 201203, China; yingchunhu@hotmail.com (Y.H.);

**Keywords:** PBPK modeling, piperine, food–drug interaction, CYP3A4

## Abstract

Piperine has been reported to inhibit the enzyme activity of cytochrome P450 (CYP) 3A4. The aim of this study was to develop and validate a physiologically based pharmacokinetic (PBPK) model for piperine and to predict potential food–drug interactions (FDIs) between piperine and CYP3A4 substrate drugs using these models. The PBPK model for piperine was successfully developed and validated. Using this model, FDIs with ten CYP3A4 substrate drugs were simulated. The predicted area under the curve (AUC) ratios (with and without piperine, following a 7-day intake of 20 mg/day) for six drugs were found to exceed 1.25, with significant increases in AUC observed for ritonavir (31%), nifedipine (34%), cyclosporine (35%), triazolam (36%), alfentanil (39%), and simvastatin (59%) in humans. These findings suggest that caution should be exercised when consuming amounts of black pepper equivalent to a daily intake of 20 mg piperine during treatment with CYP3A4 substrate drugs, as it may significantly alter their pharmacokinetics.

## 1. Introduction

Piperine (Figure 1), an alkaloid, is primarily isolated from the fruit of the Piperaceae family, including black pepper and long pepper. The average piperine content in pepper is commonly assumed to be 6% by dry weight, and the daily consumption of pepper ranges from approximately 83 to 333 mg. Consequently, doses of 5 to 20 mg of piperine are often administered daily in clinical trials [1]. In human studies, piperine has demonstrated a variety of pharmacological effects [2], such as gastrointestinal stimulation, anti-asthmatic, antioxidant, anti-hyperlipidemic, anti-diabetic, and anti-inflammatory activities, and enhancement of food absorption. Recently, piperine has garnered attention due to its ability to enhance the bioavailability of therapeutic drugs [3,4,5,6]. For instance, a dose of 20 mg/day of piperine administered for 10 consecutive days increased the AUC of carbamazepine by 47% compared to carbamazepine alone in healthy volunteers [3]. Pretreatment with piperine at 15 mg/day for three consecutive days elevated the AUC_0–5h_ of midazolam by 20% when compared to midazolam administered without piperine in clinical healthy subjects [4]. Additionally, 20 mg of pure piperine increased the serum AUC_0–3h_ of curcumin by 20-fold relative to curcumin taken without piperine [5]. It was further estimated, using a mechanistic static model, that piperine could increase the AUC of carbamazepine by 11% in humans [6]. The bioavailability enhancement of these drugs has been attributed to piperine-mediated inhibition of the CYP3A4 enzyme, as studies have reported that piperine inhibits CYP3A4 activity [4,7,8,9]. Additionally, prolonged use of piperine (20 mg/day for 14 days) was shown to suppress both mRNA and protein expression of *rCyp3a* (rat CYP3A) in Sprague Dawley rats [6].

Drug–drug interactions (DDIs) can result in reduced therapeutic efficacy or increased adverse effects, and FDIs may occur similarly. Two well-known examples are grapefruit juice and St. John’s wort. Grapefruit juice is known to increase plasma concentrations of various drugs by inhibiting the activity of CYP3A4 [10,11,12], primarily in the intestine and, to a lesser extent, in the liver, as well as by inhibiting P-glycoprotein [10,11,12,13,14], which can lead to a higher risk of clinical toxicity. Conversely, St. John’s wort (*Hypericum perforatum*), a dietary supplement and herbal remedy used for treating nervous system disorders such as depression, induces CYP3A4 activity [15,16,17]. This induction results in altered pharmacokinetics (PK) of multiple drugs, reducing plasma concentrations and diminishing clinical efficacy [18,19,20]. Similarly, pepper has long been utilized in both culinary and medicinal applications for the treatment of conditions such as influenza, fever, and digestive and respiratory disorders in traditional Chinese medicine [21] and Ayurvedic medicine [22]. The interaction between piperine, a key component of pepper, and drugs is of particular interest, as piperine has been reported to inhibit CYP3A4 activity [4,7,8,9].

Over the past few decades, the prediction of PK and DDIs using model-based approaches, incorporating physiochemical properties alongside in vitro and in vivo data, has seen significant advancement. Initially relying on static and dynamic models, these efforts have evolved toward PBPK models, which simulate time-dependent substrate and perpetrator kinetics at the interaction sites. As the field progressed, several prototyping software platforms, such as GastroPlus^®^ (Simulations Plus, Lancaster, CA, USA) and Simcyp^®^ (Certara Simcyp, Sheffield, UK), were developed to facilitate clinical PK simulations and predict DDIs by integrating physiochemical properties, in vitro/in vivo data, and clinical information. GastroPlus^®^ [23,24,25] was originally developed to predict drug absorption and is noted for its high predictive accuracy of PK profiles and parameter estimates. Therefore, it is well suited for biopharmaceutical studies focused on drug formulation and absorption. In contrast, the Simcyp^®^ PBPK simulator [23,24,25] was originally developed to study DDIs in humans. This was particularly challenging because predicting clearance in DDI scenarios is more complex than drug absorption. Today, Simcyp^®^ is one of the leading platforms for model-based drug development. PBPK models typically consist of multiple compartments that correspond to different body tissues, each defined by specific characteristics such as tissue volume, blood perfusion rates, enzyme/transporter expression levels, and tissue–plasma partition coefficients. PK parameters, along with absorption, distribution, metabolism, and excretion (ADME) properties, have been recognized as critical factors influencing the success or failure of drug candidates. Given the increasing importance of DDI studies, regulatory guidelines have been issued by the U.S. Food and Drug Administration [26,27,28], the European Medicines Agency [29], and, more recently, the National Medical Products Administration in China [30]. Understanding the PK properties and interaction potential of investigational drugs is essential in drug discovery and development. Moreover, these interactions may involve not only other drugs, such as over-the-counter medications, but also constituents from dietary sources, leading to FDIs, which are as critical as DDIs in clinical contexts.

The objective of this study was to develop and optimize PBPK modeling approaches to simulate the PK profiles of piperine and predict its potential FDIs. The Simcyp^®^ platform was employed to integrate in silico, in vitro, and in vivo data, providing a comprehensive simulation of piperine’s PK and its interactions with CYP3A4 substrate drugs (Figure 2).

## 2. Results

### 2.1. CL_int_ of Piperine in HLM

The in vitro intrinsic clearance (CL_int_) of piperine in human liver microsomes (HLMs) was measured and calculated in our laboratory (Figure 3), and previously reported values from the literature are summarized in Table 1. Our in-house CL_int_ values for piperine were 33.2 ± 1.1, 12.9 ± 0.3, 34.0 ± 1.3, and 10.6 ± 0.9 μL/min/mg protein under the following conditions: (1) HLM 0.33 mg/mL and piperine 0.1 μM, (2) HLM 0.33 mg/mL and piperine 1 μM, (3) HLM 0.5 mg/mL and piperine 0.1 μM, and (4) HLM 0.5 mg/mL and piperine 1 μM. Reported values in the literature include 10.3 μL/min/mg protein from Zabela et al. [31] and 44.2 μL/min/mg protein from Gou et al. [32], both measured under the same conditions of HLM 0.5 mg/mL and piperine 1 μM.

### 2.2. PBPK Model for Piperine

As shown in Table 2, the optimized input parameters for the PBPK model of piperine are presented. The effective permeability (P_eff_) of piperine was predicted using the Advanced Dissolution, Absorption, and Metabolism (ADAM) absorption model, based on an apparent permeability (P_app_) of 47.8 × 10^−^^6^ cm/s in the Caco-2 cell line [33] and solubility data of 0.004 mg/mL [34]. A volume of distribution at steady state (Vss) of 0.826 L/kg for humans was predicted by the Minimal PBPK Distribution Model using the default mechanistic method 2. Elimination data were obtained using the whole-organ metabolic clearance prediction method, with an average CL_int_ value of 24.2 μL/min/mg protein, calculated from both in-house and reported literature values [31,32] (Table 1). Interaction parameters, including an apparent inhibition constant (K_app_) of 6.74 μM, inactivation rate constant (k_inact_) of 0.558 1/h, and fraction unbound in microsomes (fu_mic_) of 0.863, were applied in the PBPK model. The model was validated using both single-dose (20 mg) and multiple-dose (20 mg/day for 7 days) oral administrations of piperine in Chinese healthy volunteers [35]. The predicted plasma concentration–time (C-T) profiles for both single and multiple doses were generated using the PBPK simulation, with comparisons to observed clinical data shown in Figure 4. As indicated in Table 3, the fold-error between the predicted and observed maximum plasma concentration (C_max_) and AUC_last_ values was 1.3 and 1.4 for the single dose simulation, and 1.0 and 1.1 for the multiple dose simulation, respectively. All values were within the acceptable prediction threshold of two-fold, demonstrating the reliability of the PBPK model.

### 2.3. PBPK Prediction for Piperine–Drug Interaction

Given that piperine has been reported as a CYP3A4 inhibitor in various studies [7,36], its interaction potential with several CYP3A4 substrate drugs was evaluated. A total of 10 CYP3A4 substrate drugs were selected for interaction studies to investigate the changes in their AUC and Cmax following oral administration, with or without a daily intake of 20 mg of piperine, over seven consecutive days. The simulation results are presented in Table 4 and Figure 5. Piperine was found to increase the AUC of carbamazepine, clarithromycin, midazolam, itraconazole, ritonavir, nifedipine, cyclosporine, triazolam, alfentanil, and simvastatin by 11%, 17%, 20%, 24%, 31%, 34%, 35%, 36%, 39%, and 59%, respectively, while enhancing the corresponding C_max_ by 3%, 7%, 11%, 13%, 27%, 20%, 18%, 20%, 25%, and 49%. The predicted AUC ratio of ritonavir, nifedipine, cyclosporine, triazolam, alfentanil, and simvastatin in the presence versus absence of piperine (following 7 consecutive days of administration) was ≥1.25, with respective ratios of 1.31, 1.34, 1.35, 1.36, 1.39, and 1.59. These findings indicate a potential risk for significant drug interactions and overdose when these drugs are co-administered with piperine.

The high SEMs observed for the AUC and C_max_ in Table 4 can be attributed to several factors inherent in the PBPK modeling approach. One key factor is the significant inter-individual variability incorporated in the virtual population simulated in SimCYP, which includes differences in age (18–65 years), gender, and genetic polymorphisms. This variability closely mimics real-world heterogeneity in drug metabolism and interactions. Furthermore, the substantial polymorphism of CYP3A4, with coefficients of variation (CVs) of 41% for enzyme abundance and 68% for turnover rate in this study’s virtual population, contributes to marked differences in drug metabolism across individuals. However, this variability does not necessarily compromise the predictive reliability of the model; rather, it reflects the intrinsic variability of drug interactions within populations.

## 3. Discussion

PBPK modeling is an effective and practical tool for predicting PK profiles and simulating FDIs, particularly when drug metabolism data and PK resources are limited or in vivo studies cannot be conducted regularly. In this study, the ADAM absorption model was employed to predict the rate and extent of piperine absorption in humans. This physiologically based transit model describes processes such as disintegration, de-aggregation, dissolution, solubility, and supersaturation/precipitation within the digestive tract. The total amount of the compound considered in the model includes both dissolved parts and drug particles [37], offering a comprehensive simulation of absorption dynamics. Since piperine was administered in capsule form in the clinical study, the ADAM model was applied, as it accommodates various formulations and provides enterocytic concentrations and amounts for both solid and dissolved drug fractions. For distribution, the Vss is a critical parameter, representing the ratio of the total drug quantity in the body to the concentration in plasma at steady state, where drug distribution remains unchanged [38]. Vss was predicted using Method 2 of the mechanistic models, which is based on the Rodgers and Rowland approach [39,40,41]. This method divides tissue water volume into intra- and extra-cellular components and includes the tissue acidic phospholipid fraction to account for ionization at the relevant compartment pH. This method was appropriate for piperine, a monoprotic base compound, given its ionization behavior. Metabolic clearance, a key elimination parameter, was calculated using the whole-organ metabolic clearance approach due to the absence of clinical data. Organ-specific intrinsic CL_int_ for the liver was obtained using in vitro to in vivo extrapolation (IVIVE) from in vitro metabolic data derived from HLM, with a CL_int_ (HLM) value of 24.2 μL/min/mg protein applied to the liver. For interaction studies, enzymatic parameters for piperine, such as K_app_, k_inact_, and fu_mic_, were utilized. The PBPK model for piperine was successfully established and validated, proving to be a valuable tool for predicting PK profiles and assessing FDI potential. This modeling approach has broad applications, including the preliminary assessment of potential compound interactions in clinical settings. To maintain accuracy, PBPK models should be updated with new mechanistic insights, such as enzyme or transporter inhibition or induction, to improve the prediction of PK profiles.

In the present piperine–drug interaction PBPK modeling studies, a total of 10 CYP3A4 victim drugs were studied for their interaction with piperine co-administration. After searching the 10 drugs from DrugBank Online (https://go.drugbank.com (accessed on 5 January 2024)), 8 drugs included indications for avoiding grapefruit product co-administration, and 2 of them, carbamazepine and cyclosporine, have demonstrations of their prevention due to both grapefruit products and St. John’s Wort, which are shown in Table 5. Grapefruit juice and St. John’s Wort were the inhibitor and inducer of CYP3A4, respectively; piperine was reported as a CYP3A4 inhibitor. Logically, the drugs subjected to grapefruit could be affected to a certain extent by piperine interactions as well.

It was stipulated in the DDI guidance that the predicted AUC ratio of a victim drug in the presence and absence of a studied compound was ≥1.25 based on static mechanistic models or dynamic mechanistic models (e.g., PBPK models); a clinical interaction study should be conducted [42]. Based on the simulation results in the present study, the predicted AUC ratio (with and without piperine treatments) of six drugs were above 1.25, listed as ritonavir, nifedipine, cyclosporine, triazolam, alfentanil, and simvastatin, as shown in Table 4.

Alfentanil and cyclosporine are classified as narrow therapeutic index (NTI) drugs [36,37], as they are highly dependent on CYP3A for clearance. NTI medications are particularly susceptible to potential interactions, and thus, therapeutic drug monitoring is often required to ensure efficacy and prevent toxicity. Elevated plasma concentrations of NTI drugs can lead to serious and potentially life-threatening events. Inhibition of CYP3A4 activity by piperine could elevate the plasma concentrations of alfentanil and cyclosporine. Therefore, it is advisable to avoid excessive pepper intake during the use of these medications, or patients should be closely monitored.

Nifedipine is commonly prescribed to reduce blood pressure and increase oxygen supply to the heart, alleviating angina by blocking voltage-gated L-type calcium channels [43]. Simvastatin, along with other statins, is considered a first-line treatment for hyperlipidemia through the inhibition of 3-hydroxy-3-methylglutaryl coenzyme A (HMG-CoA) reductase. Statins are standard practice following cardiovascular events to reduce mortality risk [44]. As hypertension and hyperlipidemia often co-occur as indicators of cardiovascular and cerebrovascular diseases, which are leading causes of morbidity globally [45], a large number of individuals take nifedipine and simvastatin daily. Since both drugs are predominantly metabolized by CYP3A4 [43,46], caution should be exercised regarding the daily consumption of pepper during their use.

The interaction between piperine and several drugs has been documented in the literature. Shoba et al. reported that after administering 2 g of pure curcumin powder alone or combined with 20 mg of pure piperine powder to 10 healthy male volunteers, the serum AUC_0–3h_ was 0.004 and 0.08 µg/mL∙h, respectively, indicating a 20-fold increase in the relative bioavailability of curcumin with piperine co-administration [5]. In a clinical study involving 12 healthy volunteers [3], a single dose of 200 mg carbamazepine was orally administered, with or without 10 consecutive days of piperine intake at 20 mg/day. The AUC of carbamazepine was 233 and 158 µg/mL∙h, respectively. This 47% enhancement in AUC was attributed to the inhibition of CYP3A4 by piperine, as oral clearance of carbamazepine was significantly reduced by 38.9%. Additionally, in a clinical study with healthy volunteers [4], the AUC_0–5h_ of midazolam for the piperine pretreatment group (15 mg p.o. for three consecutive days before midazolam, followed by midazolam 10 mg p.o.) and the control group (midazolam 10 mg p.o.) were 495.9 and 411.1 ng/mL∙h, respectively. The ratio of AUC_0–5h_ for midazolam was 1.21, which closely matched the simulated result of 1.2 for midazolam 5 mg co-administered with piperine for 7 consecutive days [4].

Thus, it is crucial to address the safety concerns regarding excessive daily consumption of pepper during the administration of CYP3A4 substrate drugs in clinical practice. Special caution is warranted when natural dietary supplements with strong CYP enzyme inhibition are used concurrently with modern therapeutic drugs.

## 4. Materials and Methods

### 4.1. Reagents

Piperine (purity ≥ 97.0%) was purchased from J & K Technology Co., Limited (Beijing, China). Pooled HLM (lot: IQF) from 150 donors (75 females and 75 males) were obtained from BioreclamationIVT (New York, NY, USA). Nicotinamide adenine dinucleotide phosphate (NADPH, lot: 1031B024) was sourced from Solarbio^®^ Life Science (Beijing, China). All other reagents and solvents used were of chromatographic or analytical grade.

### 4.2. CL_int_ of Piperine

The CL_int_ of piperine using HLM was determined through an in vitro incubation method, with each experiment performed in triplicate. The total incubation volume was 450 µL, and the incubation was carried out at 37 °C for 60 min with shaking at 450 rpm. The incubation mixture comprised the test compound piperine (0.1 or 1 µM) with a co-solvent of 0.01% DMSO and 0.005% bovine serum albumin (BSA), Vivid^®^ (2 µM), HLM (0.33 or 0.5 mg/mL microsomal protein), co-factor MgCl_2_ (5 mM), NADPH (1 mM), and 0.1 M Tris buffer (pH 7.4). The mixture was pre-incubated at 37 °C for 10 min with stirring, and the reactions were initiated by adding NADPH. Aliquots (50 µL) of the incubation medium were sampled at 0, 7, 17, 30, and 60 min and transferred to a cooled (+4 °C) 384-well plate containing 50 µL of methanol to terminate the enzymatic reactions.

After shaking, the fluorescence of Vivid^®^ was measured (excitation at 420 nm, emission at 465 nm, gain 60) to verify the integrity of the experimental process. Absolute zero treatments with deactivated HLM were used as controls to monitor sample collection at time zero. Vivid^®^ served as a positive control, and the fluorescent product obtained after incubation was measured to ensure accuracy. Only when the fluorescent values of Vivid^®^ production met laboratory standards did the study proceed. After centrifugation, the supernatant was collected for quantitative analysis of piperine using liquid chromatography with tandem mass spectrometry (LC-MS/MS).

The remaining percentage of the substrate was calculated using the zero-time substrate concentration as 100%. The slope was derived from the linear regression of the natural logarithm of the remaining piperine percentage against incubation time. As per the literature [47], the CL_int_ (mL/min/g protein) of piperine was calculated using Formula (1).
(1)$${\mathrm{CL}}_{\mathrm{int}}(\mathrm{mL}/\min/g\;\mathrm{protein})=\frac{1000\;\times \;\mathrm{slope}}{P}$$
where slope is the slope of the natural logarithmic remaining percentage against incubation time, and P is the final protein concentration of HLM (mg/mL).

### 4.3. LC-MS/MS Method

The concentrations of piperine in samples were measured using a Waters ACQUITY I-Class System (Waters, Milford, MA, USA) coupled with the Waters ACQUITY XEVO TQ-S (Waters, Milford, MA, USA) equipped with an electrospray ionization source operated in positive ion mode. Chromatographic separation was performed on an analytical column of ACQUITY UPLC HSS T3 (50 mm × 2.1 mm I.D., 1.8 μm) (Waters, Milford, MA, USA). The separation mobile phase consisted of 0.1% (*v*/*v*) formic acid in water (A) and 0.1% (*v*/*v*) formic acid in acetonitrile–methanol (9:1, *v*/*v*) (B). The gradient started at 1 min with 20% of B and increased to 90% of B within 3 min, followed by a column washing step with 100% of B for 1 min. The column temperature was 45 °C with a flow rate of 0.5 mL/min. The multiple reaction monitoring mode (MRM) was chosen to acquire the quantitative data. The ion source temperature was set at 500 °C, and the ion spray voltage was set at 5000 V. The ion transition for piperine was *m*/*z* 286 → *m*/*z* 115. The de-clustering potential and collision energy were 22 V and 30 V. Data were acquired with Masslynx 1.4.1. 

### 4.4. PBPK Modeling Program

All PBPK modeling was performed using the Simcyp^®^ simulator (Version 18). The process of PBPK modeling included parameters input, optimization, and validation with clinical data. All the simulations used a virtual population in the trial design with a people size of 100, including 10 trials and 10 subjects in each trial. 

The physiochemical and ADME properties of piperine were collected from SciFinder or predicted with the Simcyp^®^ simulator. With respect to the parameter inputs, several parts were needed to be written, including Phys Chem and Blood Binding, Absorption, Distribution, Elimination, and Interaction. Several available in silico and in vitro data, including molecular weight (g/mol), log *P*, compound type, acid dissociation constant (pKa), B/P ratio, and f_u,plasma_, were input in the box of Phys Chem and Blood Binding; for the Absorption part, fu_gut_, Q_gut_ (L/h), and P_eff,man_ (10^−4^ cm/s) were predicted via permeability assays using the chosen ADAM absorption model; Vss (L/kg) was input from prediction via drug-specific physiochemical parameters using mathematical formula Method 2 which was published by Rodgers and Rowland [39,40,41]; CL_int_ (HLM) (μL/min/mg protein) data were obtained via IVIVE from in vitro metabolic data in an Elimination model. Each organ was assumed to be perfusion rate limited, and the liver and kidney were considered to be the only organs to eliminate the compounds in this study.

Plasma C-T profiles of piperine were obtained from published clinical studies [35] using the Getdata 2.20 software and used as optimization and verification comparators for the PBPK modeling. 

### 4.5. Compounds and PBPK Modeling

The in silico prediction of physiochemical properties for piperine was estimated from the website of SciFinder using Advanced Chemistry Development (ACD/Labs) software V11.02, absorption and distribution parameters were predicted with the Simcyp simulator, the elimination input was extrapolated from in vitro data which was obtained from the literature and in-house metabolic stability results. A PBPK model was set up in humans, followed by optimization and validation with the observed PK data. 

The prediction accuracy of PBPK models were assessed according to the fold-error between the observed value and the predicted value, which was also widely accepted [48,49]. The fold-error was calculated using the following Formula (2):(2)$$\mathrm{Fold}-\mathrm{error}=\frac{\mathrm{Predicted}\;\mathrm{value}}{\mathrm{Observed}\;\mathrm{value}}$$

A two-fold error range (0.5–2.0) was used as a criterion to evaluate the predicted performance for PK parameters [50,51,52]. This two-fold criterion has been employed in several pediatric population PBPK models [53,54]. Although some researchers have used broader (3-fold) [55] or narrower (1.5-fold) error ranges [56], the two-fold error range remains the most commonly employed [57,58,59]. Therefore, in this study, the two-fold error range was considered as a reliable predictor.

### 4.6. Interaction Modeling and Simulation

Interaction modeling for piperine and drugs was conducted using the dynamic simulation in Simcyp. The validated PBPK model was employed for piperine–drug interaction prediction by considering the inhibitory effect of piperine as a perpetrator on CYP3A4 victim drugs which were applied in clinical studies. The interaction parameters for piperine including k_inact_ and K_app_ on the activity of CYP3A4 were obtained from the literature [6]. The information and optimized input data of CYP3A4 victim drugs were applied using Simcyp built-in models. In total, 10 drugs, namely carbamazepine, clarithromycin, midazolam, itraconazole, ritonavir, nifedipine, cyclosporine, triazolam, alfentanil, and simvastatin, were selected for interaction simulation.

## 5. Conclusions

The PBPK model for piperine was successfully developed and validated, and simulations of FDIs with CYP3A4 substrate drugs were conducted with or without a consecutive 7-day intake of 20 mg/day piperine using the validated PBPK model along with Simcyp’s built-in drug models. The results showed significant AUC enhancement for several drugs, including ritonavir, nifedipine, cyclosporine, triazolam, alfentanil, and simvastatin, indicating that caution should be exercised regarding excessive consumption of pepper during treatment with these medications. This study demonstrated that PK parameters and potential FDIs can be reliably predicted using a well-constructed PBPK model that incorporates in silico, in vitro, and in vivo data. It is recommended that PBPK modeling be routinely employed for the prediction of FDIs in studies involving drugs and active food ingredients.

## Figures and Tables

**Figure 1 ijms-25-10955-f001:**
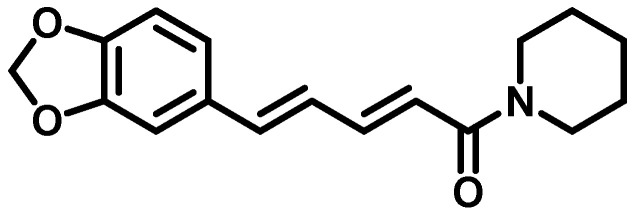
Chemical structure of piperine.

**Figure 2 ijms-25-10955-f002:**
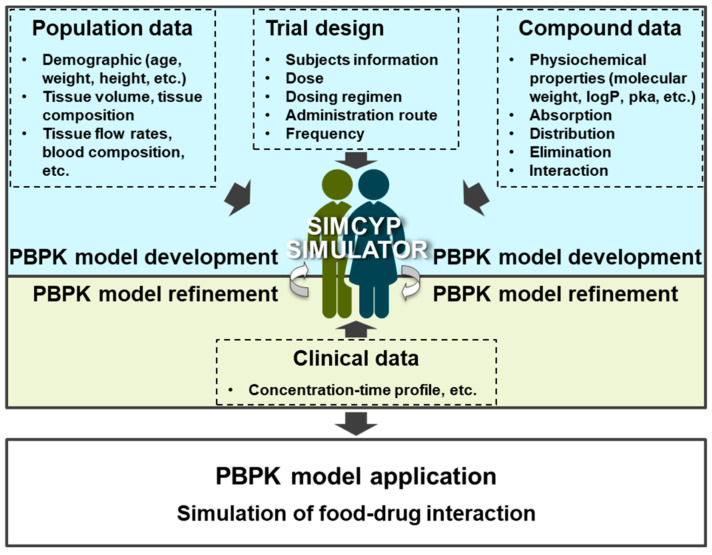
Workflow for development, optimization, and evaluation of PBPK models.

**Figure 3 ijms-25-10955-f003:**
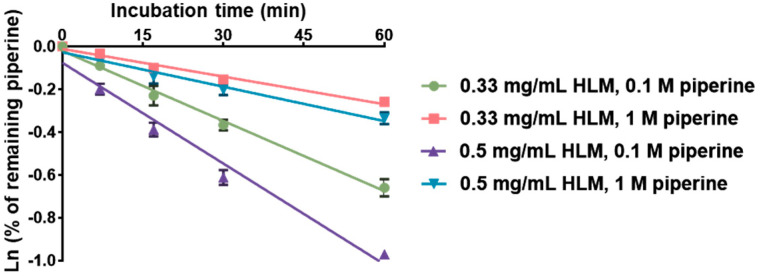
The linear regression of natural logarithmic remaining percentage of piperine against incubation time. Each value is the mean of three measurements.

**Figure 4 ijms-25-10955-f004:**
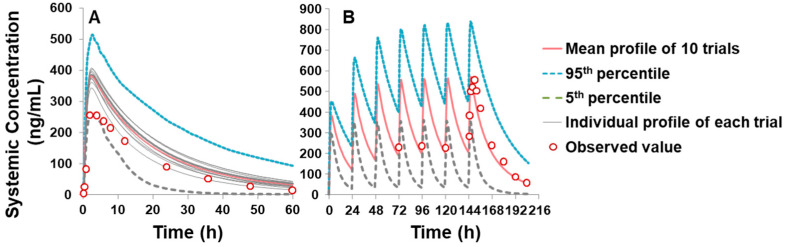
Simulated and observed piperine plasma C-T profile in Chinese healthy volunteers: (**A**) single dose (20 mg) and (**B**) multiple dose (consecutive 7 days of piperine at 20 mg/day). The 10 trial profiles are displayed as a grey line, and the mean value of simulations are presented as a black line, 5% and 95% percentile are shown as a dashed line. Open circles indicate observed values reported by Wang et al. [35].

**Figure 5 ijms-25-10955-f005:**
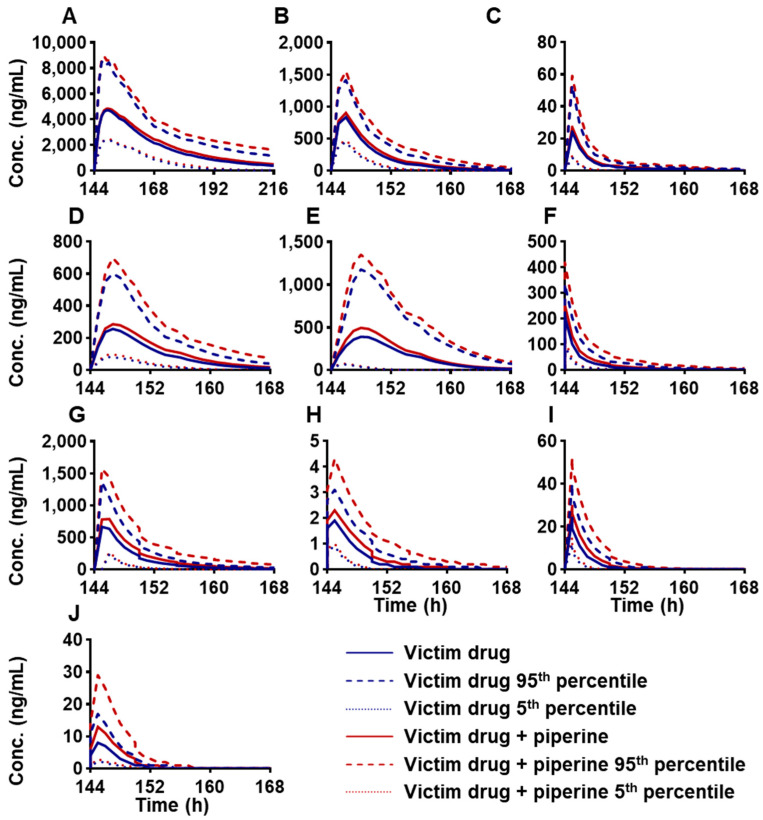
FDI prediction of plasma C-T profiles of oral administration of CYP3A4 victim drugs: (**A**) carbamazepine, 400 mg, (**B**) clarithromycin, 250 mg, (**C**) midazolam, 5 mg, (**D**) itraconazole_fed capsule, 200 mg, (**E**) ritonavir, 100 mg, (**F**) nifedipine, 20 mg, (**G**) cyclosporine, 200 mg, (**H**) triazolam, 0.25 mg, (**I**) alfentanil, 0.043 mg, (**J**) simvastatin, 40 mg, with or without consecutive 7 days of piperine at 20 mg/day in healthy volunteers. The mean value of 10 trial simulations are displayed, with the single dose of victim drugs as a blue line and victim drugs with a consecutive 7 days of piperine at 20 mg/day as a red line; 5% and 95% percentile are shown as a dashed line.

**Table 1 ijms-25-10955-t001:** In vitro CL_int_ data (Mean ± SD) of piperine obtained in-house and from the literature.

CL_int_, HLM (μL/min/mg Protein)	Concentration of HLM (mg/mL)	Concentration of Piperine (μM)	Data Source (Reference)
33.2 ± 1.1	0.33	0.1	In-house
12.9 ± 0.3	0.33	1	In-house
34.0 ± 1.3	0.5	0.1	In-house
10.6 ± 0.9	0.5	1	In-house
10.3	0.5	1	Zabela et al. [31]
44.2	0.5	1	Gou et al. [32]

**Table 2 ijms-25-10955-t002:** Summary of piperine input parameters in Simcyp.

Item	Parameter	Value	Source/Reference
Physiochemical properties	MW (g/mol)	285.34	SciFinder
	log P	1.86	Fu et al. [34]
	Compound type	Monoprotic base	SciFinder
	pKa	12.22	Fu et al. [34]
	B/P ratio	0.63	Fu et al. [34]
	f_u,plasma_	0.03	Ren et al. [33]
Absorption	Absorption Model-ADAM Model		
	fu_gut_	0.0211	Simcyp predicted
	Q_gut_ (L/h)	15.4	Simcyp predicted
	P_eff,man_ (10^−4^ cm/s)	5.13	Simcyp predicted
	Permeability Assay	Caco-2	
	P_app_ (10^−6^ cm/s)	47.8	Ren et al. [33]
	Solubility (mg/mL)	0.004	Fu et al. [34]
Distribution	Distribution Model-Minimal PBPK Model		
	V_SS_ (L/kg)	0.826	Predicted-Method 2
Elimination	Clearance type-Whole Organ Metabolic Clearance		
	CL_int_, HLM (μL/min/mg protein)	24.2	Publications [31,32] and in-house data
Interaction	Enzyme	CYP3A4	
	K_app_ (μM)	6.74	Ren et al. [33]
	k_inact_ (1/h)	0.558	Ren et al. [33]
	fu_mic_	0.863	Zabela et al. [31]
Clinical data	Wang et al. [35]		

**Table 3 ijms-25-10955-t003:** Results on fold-error of C_max_ and AUC between predicted and observed parameter values of piperine.

Administration	Parameter	Predicted Value ^1^	Observed Value	Fold-Error
Single dose 20 mg	C_max_ (ng/mL)	382 ± 83	290 ± 15	1.3
	AUC_last_ (ng/mL∙h)	7929 ± 298	5642 ± 338	1.4
Multiple dose 20 mg/day, 7 days	C_max_ (ng/mL)	571 ± 12	595 ± 39	1.0
	AUC_last_ (ng/mL∙h)	12,800 ± 397	14,356 ± 1365	1.1

^1^ Predicted value (Mean ± SEM (Standard Error of the Mean)) came from Simcyp software (Version 18) prediction based on input data and observed value from clinical data [35].

**Table 4 ijms-25-10955-t004:** Interaction prediction of CYP3A4 victim drugs with and without consecutive 7 days of piperine via oral administration at 20 mg/day.

Victim Drug		Treatment	Geometric Mean ± SEM			
Name	Dose (mg)		AUC (ng/mL∙h)	C_max_ (ng/mL)	Ratio of AUC	Ratio of C_max_
Carbamazepine	400	SV-Carbamazepine	122,486 ± 4691	4480 ± 191	1.11	1.03
		SV-Carbamazepine + Piperine	135,775 ± 5354	4592 ± 197		
Clarithromycin	250	SV-Clarithromycin	4298 ± 203	844 ± 33	1.17	1.07
		SV-Clarithromycin + Piperine	5034 ± 243	905 ± 35		
Midazolam	5	Sim-Midazolam	62.9 ± 4.3	21.6 ± 1.4	1.20	1.11
		Sim-Midazolam + Piperine	75.8 ± 5.3	23.9 ± 1.6		
Itraconazole	200	SV-Itraconazole_Fed Capsule	2050 ± 164	224 ± 15	1.24	1.13
		SV-Itraconazole_Fed Capsule + Piperine	2533 ± 198	253 ± 17		
Ritonavir	100	SV-Ritonavir	2168 ± 401	285 ± 37	1.31	1.27
		SV-Ritonavir + Piperine	2846 ± 462	361 ± 43		
Nifedipine	20	Sim-Nifedipine	381 ± 22	208 ± 7	1.34	1.20
		Sim-Nifedipine + Piperine	512 ± 35	249 ± 9		
Cyclosporine	200	SV-Cyclosporine	2606 ± 173	765 ± 32	1.35	1.18
		SV-Cyclosporine + Piperine	3520 ± 239	906 ± 37		
Triazolam	0.25	SV-Triazolam	6.72 ± 0.43	1.87 ± 0.08	1.36	1.20
		SV-Triazolam + Piperine	9.13 ± 0.62	2.24 ± 0.10		
Alfentanil	0.043	SV-Alfentanil	50.6 ± 2.1	23.5 ± 0.9	1.39	1.25
		SV-Alfentanil + Piperine	70.3 ± 2.4	29.4 ± 0.1		
Simvastatin	40	SV-Simvastatin	22.9 ± 1.7	6.89 ± 0.51	1.59	1.49
		SV-Simvastatin + Piperine	36.5 ± 3.5	10.3 ± 0.9		

**Table 5 ijms-25-10955-t005:** Food interactions of 10 CYP3A4 victim drugs from DrugBank Online.

Name	Food Interactions	Source (Accessed on 5 January 2024)
Carbamazepine	Avoid grapefruit products. Avoid St. John’s Wort.	https://go.drugbank.com/drugs/DB00564
Clarithromycin	No interactions found.	https://go.drugbank.com/drugs/DB01211
Midazolam	Avoid grapefruit products.	https://go.drugbank.com/drugs/DB00683
Itraconazole	Avoid grapefruit products.	https://go.drugbank.com/drugs/DB01167
Ritonavir	Avoid St. John’s Wort.	https://go.drugbank.com/drugs/DB00503
Nifedipine	Avoid grapefruit products.	https://go.drugbank.com/drugs/DB01115
Cyclosporine	Avoid grapefruit products. Avoid St. John’s Wort.	https://go.drugbank.com/drugs/DB00091
Triazolam	Avoid grapefruit products.	https://go.drugbank.com/drugs/DB00897
Alfentanil	No interactions found.	https://go.drugbank.com/drugs/DB00802
Simvastatin	Avoid grapefruit products.	https://go.drugbank.com/drugs/DB00641

## Data Availability

The original contributions presented in the study are included in the article, further inquiries can be directed to the corresponding author.

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
