# Peer review of "Predicting Food–Drug Interactions between Piperine and CYP3A4 Substrate Drugs Using PBPK Modeling"

_ijms, 2024, doi:10.3390/ijms252010955_

Round 1

Reviewer 1 Report

Comments and Suggestions for Authors

This a very organized and well written manuscript. The scientific approach is sound. The authors analyzed the content and intrinsic clearance of piperine, used these values in PBPK model using SymCip software, and validated the predictions based on previous publications from other authors. The text is fluent. The figures and tables are informative and of good graphical quality. The English is fine and no additional proofing is needed. I have only 2 small comments related to the contents of this manuscript:

The Abstract and other locations in the text – “cyclosporine_neoral” – please replace with “cyclosporine”. Provide explanations in the Methods, if necessary.

Line 209 – definition of the NTI drug is not related to administration with or without cannabinoids or any other compounds. Please rephrase.

Author Response

Comments 1. The Abstract and other locations in the text – “cyclosporine_neoral” – please replace with “cyclosporine”. Provide explanations in the Methods, if necessary.

Response 1: The term "cyclosporine_neoral" in the full text has been replaced with "cyclosporine".

Comments 2. Line 209 – definition of the NTI drug is not related to administration with or without cannabinoids or any other compounds. Please rephrase.

Response 2: The definition of alfentanil and cyclosporine as the NTI drugs has been rephrased.

Reviewer 2 Report

Comments and Suggestions for Authors

I have read this paper with a background on clinical pharmacology and DDI/food drug interactions, with experience in the use and development of PBPK models, without being a 'hardcore' PBPK modeller. I assume that other reviewers will further add to this specific skills. 

I value the work, that further explores a well established pathway of food-drug interaction, that has up to regulatory approval for labelling of drugs. Consequently, the methods are not really new on the approach taken and pathway, while the findings and results on the piperine are indeed new. It therefore rather reads as an new application of a known pathway (as also mentioned by the authors, cfr other food-drug examples on CYP3A4)

I only have specific comments

the authors should be much clearer on the route of administration tested (cf alfentanil) throughout the paper. 

in the abstract, 'large amounts of pepper' does not really provide accurate information, i highly recommend to quantify this. 

The potential mechanism is mentioned in lines -, but how does this fit with the rather short in time in vitro experiments ? 

I would recommend to provide a broader overview of software platforms (line 65-67).

Please check figure 1: i think that there are typo's overthere: simcyp simulator (instead of simulaor), and physiochemical or physicochemical ? 

methods, line 324

the fold error <2 is commonly used, while quite broad (versus more stringent error ranges), i would suggest to add a reference on this, and further reflect on this in the discussion. 

Author Response

Comments 1. The authors should be much clearer on the route of administration tested (cf alfentanil) throughout the paper.

Response 1: The route of administration was described in Lines 145-148.

Comments 2. In the abstract, 'large amounts of pepper' does not really provide accurate information, I highly recommend to quantify this.

Response 2: The information of equivalent intake of piperine has been added.

Comments 3. The potential mechanism is mentioned in lines -, but how does this fit with the rather short in time in vitro experiments?

Response 3: In this study, we used a PBPK model to predict in vivo DDIs. Despite the short incubation time in vitro inhibition experiments, the PBPK model can more comprehensively reflect the dynamic changes of drugs in vivo by integrating various in vivo and in vitro data. Specifically, we considered the following aspects in the model:

  1. Adjustment of Kinetic Parameters: We appropriately corrected and scaled the inhibition constants (such as hepatic clearance) obtained from in vitro experiments to suit in vivo conditions. This includes considering factors like plasma protein binding rates and tissue distribution to address differences caused by the short incubation time.
  2. Time-Dependent Effects: The PBPK model can simulate the time-concentration curve in vivo. Thus, even with short incubation times in in vitro inhibition experiments, the model can predict DDIs that better align with actual in vivo situations by simulating concentration changes at different time points.
  3. Integration of Multiple Data Sources: In addition to in vitro inhibition data, we also integrated clinical pharmacokinetic data and other relevant in vivo data to enhance the accuracy and reliability of the model's predictions. This integration of multiple data sources helps mitigate the limitations of single in vitro experiments.

In summary, despite the short incubation time in in vitro inhibition experiments, the PBPK model can effectively match in vitro data with in vivo prediction results through the methods described above, thereby providing reliable DDI predictions.

Comments 4. I would recommend to provide a broader overview of software platforms (line 65-67).

Response 4: The overview of software platforms was added in the original paragraph.

Comments 5. Please check figure 1: I think that there are typo's overthere: simcyp simulator (instead of simulaor), and physiochemical or physicochemical?

Response 5: “simulator” in Figure 2 has been corrected. “Physicochemical” in the full text has been replaced with “physiochemical”.

Comments 6. Methods, line 324, the fold error <2 is commonly used, while quite broad (versus more stringent error ranges), I would suggest to add a reference on this, and further reflect on this in the discussion.

Response 6: More discussion has been added in section 4.5.

Reviewer 3 Report

Comments and Suggestions for Authors

ijms-3239821

Predicting Food-Drug Interactions Between Piperine and CYP3A4 Substrate Drugs Using PBPK Modeling

The manuscript by Lin et al. described the development and evaluation of a PBPK model for piperine to predict its potential food-drug interactions with CYP3A4 substrate drugs. The manuscript was well prepared and the data were sufficient for the conclusion. Below are some specific comments. Please consider them to improve the manuscript.

1. LC-MS/MS method: “The ion transition for piperine was m/z 115 → m/z 286”. This is incorrect. Please recheck. Also, please verify the ion transition considering the drug structure and include it in the Results and Discussion.

2. Figure 3:  Please use colors to distinguish different conditions.

3. Table 1: Please include SD or SE.

4. Figure 4: Please use colors to distinguish the Mean profile, 95th percentile, and 5th percentile.

5. Table 3: Please include SD or SE to observed values if available.

6. Table 4: Please include SD or SE.

7. Figure 5: Please increase the size of each graph for clarity.

Author Response

Comments 1. LC-MS/MS method: “The ion transition for piperine was m/z 115 → m/z 286”. This is incorrect. Please recheck. Also, please verify the ion transition considering the drug structure and include it in the Results and Discussion.

Response 1: The ion transition was corrected as m/z 286 → m/z 115.

Comments 2. Please use colors to distinguish different conditions.

Response 2: The different conditions in figure 3 have been colored.

Comments 3. Table 1: Please include SD or SE.

Response 3: The SD values have been added in Table 1.

Comments 4. Figure 4: Please use colors to distinguish the Mean profile, 95th percentile, and 5th percentile.

Response 4: Revised accordingly (Figure 4).

Comments 5. Table 3: Please include SD or SE to observed values if available.

Response 5: Revised accordingly (Table 3).

Comments 6. Table 4: Please include SD or SE.

Response 6: Revised accordingly (Table 4).

Comments 7. Figure 5: Please increase the size of each graph for clarity.

Response 7: Revised accordingly (Figure 5).

Round 2

Reviewer 2 Report

Comments and Suggestions for Authors

i agree with the rebuttal and the revision

Author Response

Thank you for your positive feedback and for agreeing with our rebuttal and revisions.

Reviewer 3 Report

Comments and Suggestions for Authors

The manuscript was appropriately revised. Below are some other points to consider in the revised manuscript.

1. Figure 4: The legend on the right should be changed accordingly.

2. Table 4: The AUC and Cmax vary substantially. In some cases, the RSD is ~200%. Please discuss this and provide explanations.

Author Response

Comments 1. Figure 4: The legend on the right should be changed accordingly.

Response 1: Revised accordingly (Figure 4).

Comments 2. Table 4: The AUC and Cmax vary substantially. In some cases, the RSD is ~200%. Please discuss this and provide explanations.

Response 2:

We appreciate the reviewer’s constructive comments. We double check the raw data and calculation process. The reasons for the high coefficients of variation (CV) for AUC and Cmax in Table 4 are summarized as follows:

  1. Inter-individual Variability: The virtual populations simulated in SimCYP incorporates inter-individual variability by including differences in age (18–65 years), gender, and genetic polymorphisms, which influence drug metabolism and interactions, resulting in variable AUC and Cmax values. This inherent variability reflects real-world observations and is characteristic of PBPK modeling.
  2. CYP3A4 Enzyme Characteristics: The substantial polymorphism of CYP3A4, with a CV of 41% for abundance and 68% for turnover rate in this study's virtual population, leads to marked differences in drug metabolism across individuals, further contributing to the variability in PK parameters.
  3. Model Assumptions and Simplifications: PBPK models inevitably incorporate assumptions and simplifications. These may introduce discrepancies between simulated and actual physiological processes in certain instances, thus contributing to the variability in predicted outcomes.

However, this variability does not necessarily compromise predictive reliability; rather, it reflects the inherent variability in drug interactions within populations. This underscores the importance of considering individual factors when evaluating the piperine-CYP3A4 substrate interaction, which has significant clinical implications.

Additionally, in this study, SEM values can more directly reflect the reliability of the average values of the simulation results, which is very helpful for assessing the overall trends of drug interactions. Therefore, the SD values in Tables 3 and 4 have been replaced with SEM values.

Round 3

Reviewer 3 Report

Comments and Suggestions for Authors

The response to comment #2 should be included in the main text where relevant.

Author Response

Comments1: The response to comment #2 should be included in the main text where relevant.

Response 1: The response has been included in Section 2.3 on Page 5.